# Correlations between Quality of Life, School Bullying, and Suicide in Adolescents with Attention-Deficit Hyperactivity Disorder

**DOI:** 10.3390/ijerph17093262

**Published:** 2020-05-07

**Authors:** Yi-Lung Chen, Hsing-Ying Ho, Ray C. Hsiao, Wei-Hsin Lu, Cheng-Fang Yen

**Affiliations:** 1Department of Healthcare Administration, Asia University, Taichung 41354, Taiwan; elong@asia.edu.tw (Y.-L.C.); hyhftc@gmail.com (H.-Y.H.); 2Department of Psychology, Asia University, Taichung 41354, Taiwan; 3Department of Psychiatry and Behavioral Sciences, University of Washington School of Medicine, Children’s Hospital, Seattle, WA 98105, USA; rhsiao@u.washington.edu; 4Department of Psychiatry, Ditmanson Medical Foundation Chia-Yi Christian Hospital, Chia-Yi City 60002, Taiwan; 5Department of Senior Citizen Service Management, Chia Nan University of Pharmacy and Science, Tainan 71710, Taiwan; 6Department of Psychiatry, Kaohsiung Medical University Hospital, Kaohsiung Medical University, Kaohsiung 80708, Taiwan; 7Department of Psychiatry, School of Medicine and Graduate Institute of Medicine, College of Medicine, Kaohsiung Medical University, Kaohsiung 80708, Taiwan

**Keywords:** quality of life, school bullying, suicide, ADHD

## Abstract

Although adolescents with attention-deficit hyperactivity disorder (ADHD) have a higher risk of suicidality and more problems related to school bullying, and quality of life (QoL) is reportedly associated with school bullying, suicide, and ADHD, no study has examined their correlation. This study examined the complex relationships between QoL, school bullying, suicide, and ADHD symptoms. A total of 203 adolescents with ADHD aged between 12 and 18 years were recruited. School bullying and QoL were examined using the Chinese version of the School Bullying Experience Questionnaire and the Taiwanese Quality of Life Questionnaire for Adolescents. Network model analysis was conducted to graphically present their relationships. We identified triangular correlations between school bullying, QoL, and suicidality, indicating possible pathways from school bullying to suicidality, and the originating or mediating roles of personal competence and psychological well-being. Furthermore, the ADHD symptoms of inattention and hyperactivity/impulsivity may differentially regulate these pathways. Longitudinal studies are warranted to confirm these findings.

## 1. Introduction

Adolescents with attention-deficit hyperactivity disorder (ADHD) have been reported to be involved in school bullying. A high prevalence of bullying involvement has been reported for these adolescents, with 14.6% being victims, 8.4% being perpetrators, and 5.6% being victim-perpetrators [1]. Bullying patterns change with age. Children with ADHD are more likely to be victims of school violence at elementary school; however, in junior high school, they become both victims and perpetrators, although school bullying problems somewhat decrease among them in high school [2]. Bullying can impair the physical and mental health of adolescents with ADHD. For instance, victims and perpetrators of bullying have more symptoms of anxiety and depression [3]. In one study, children and adolescents who were victims and perpetrators of bullying reported higher self-perceived pain in the chest, abdomen, neck and shoulders, back and extremities, and head than those who did not report involvement in bullying, and the pain enhanced the severity of depression, anxiety, and sleep problems [4]. Furthermore, the bullying experiences of children with ADHD might mediate psychotic experiences in later life [3]. Therefore, understanding the impact of bullying on adolescents with ADHD can help decrease their impairments later in life.

Bullying has been reported as a risk factor for suicide in adolescents, contributing to 6.4% of all causes of suicide deaths [5]. Victimization from bullying is associated with suicide attempts across countries, and the victimization frequency is correlated with suicidal ideation, suicide attempts, and psychiatric disorders, including depression [6,7]. Different types of bullying have different consequences for suicide among young victims. For example, in a cross-sectional study, physical bullying and verbal bullying caused a higher risk of suicidal ideation, whereas relational bullying contributed to a higher risk of suicide attempts [8]. Similar results regarding victimization have been found in longitudinal studies, in which researchers reported that bullying victimization at an early age could predict suicidal ideation and suicide attempts in adolescents [9,10]. Additionally, both bullying victimization and perpetration were related to a higher frequency of suicide attempts among adolescent inpatients in an acute inpatient program [11]. Hence, clarifying the relationship between bullying and suicide is pivotal for reducing suicidal problems among adolescents.

Evidence suggests that individuals with ADHD are at a higher risk of developing suicide-related behaviors, including suicidal ideation, suicide plans, suicide attempts, and suicide death [12]. Children with ADHD are three times more at risk of suicidal ideation, suicide plans, and suicide attempts than children without ADHD after adjusting for the effects of family dysfunction and comorbid psychiatric conditions [13]. In a long-term observation, adolescents with ADHD were at a higher risk of suicidality and repeated suicide attempts in adulthood with or without a psychiatric disorder comorbidity [14,15]. Although psychological problems concurrent with ADHD, such as depressive symptoms, specific anxiety disorders, dysthymia symptoms [13,16,17], weak cognitive function [18], and family dysfunction [13], are commonly reported as being involved in the mechanisms of suicide, a pooled data analysis of suicidality and ADHD demonstrated that ADHD is an independent risk factor for suicidality [12].

Quality of life (QoL) is defined by the World Health Organization as an individual’s perception of their position in life in the contexts of the cultures and value systems in which they live in relation to their goals, expectations, standards, and concerns [19]. The protective effects of QoL have been reported for several diseases. For instance, higher health-related QoL was associated with accelerated adaptive behaviors in adolescents with anemia [20]. Similarly, higher health-related QoL can predict higher survival rates for many diseases, such as hepatic carcinomas [21], heart disease [22], and severe pulmonary diseases [23,24,25]. Moreover, the protective roles of QoL on depression were observed in caregivers of partners with chronic disorders as well as the elderly population [26,27], suggesting QoL not only affected physical health or behaviors, but also mental health. However, no study has examined the associations between QoL, mental health, and school bullying simultaneously in adolescents with ADHD.

In this study, we examined the associations between school bullying, QoL, and suicidality in adolescents with ADHD. We recruited 203 adolescents who had received a clinical diagnosis of ADHD and measured their school bullying involvement, suicidality, and QoL through self-reported questionnaires. We then applied network analysis to examine the associations among the victimization and perpetration of physical bullying, belongings snatching, verbal and relational bullying, QoL, and suicidality. Based on the aforementioned literature, we hypothesized that adolescents with experiences of school bullying would report lower QoL, and that a negative association would be observed between QoL and suicidality. We also examined the role of ADHD symptoms in these associations.

## 2. Materials and Methods

### 2.1. Participants

Study participants were recruited from child and adolescent psychiatric outpatient clinics in Kaohsiung and Taipei, Taiwan. Adolescents who were aged between 12 and 18 years, and who were diagnosed as having ADHD according to the criteria in the *Diagnostic and Statistical Manual of Mental Disorders (DSM), Fourth Edition, Text Revision* [28], were consecutively invited to participate in this study between November 2009 and July 2012. ADHD was diagnosed according to an interview of the participants’ parents with a child psychiatrist using the ADHD module of the Mini-International Neuropsychiatric Interview for Children and Adolescents [29]. Adolescents who exhibited intellectual disability, schizophrenia, bipolar disorder, pervasive developmental disorder, difficulty in verbal communication, or cognitive deficits that prevented them from understanding the study purpose or completing the questionnaires were excluded. Originally, a total of 579 children and adolescents diagnosed as having ADHD were invited to participate. Among them, 474 (81.9%) agreed to participate and were interviewed by research assistants using the study questionnaire. A study based on the same data has been published [4]. In this study, we included participants aged ≥12 years because those younger than 12 did not complete the assessment of QoL, one of the variables of interest. After excluding these participants, a total of 203 adolescents who were both diagnosed with ADHD and aged ≥12 years were included. No significant differences were identified in gender (*p* > 0.05) or age (*p* > 0.05) between those who agreed and those who refused to participate.

### 2.2. Ethics

The study was performed in accordance with the Declaration of Helsinki. The Institutional Review Boards of Kaohsiung Medical University Hospital, Chang Gung Memorial Hospital, and Kaohsiung Medical Center approved the study. All participants and their parents provided written informed consent before completing the questionnaires.

### 2.3. Measures

#### 2.3.1. Chinese Version of the Swanson, Nolan, and Pelham, Version IV

The Swanson, Nolan, and Pelham, Version IV (SNAP-IV) employs a 4-point Likert scale with a score of 0 (not at all), 1 (just a little), 2 (quite a bit), and 3 (very much). It contains 18 items according to the core symptoms identified in the DSM-IV for symptoms of inattention-deficit (items 1–9) and hyperactivity/impulsivity (items 10–18) [30]. We used the Chinese version of SNAP-IV parent forms to assess the ADHD-related symptoms of participants. The version has demonstrated satisfactory test-retest reliability, internal consistency, and concurrent and discriminant validity [31].

#### 2.3.2. Chinese Version of the School Bullying Experience Questionnaire

The Chinese version of the self-reported School Bullying Experience Questionnaire (C-SBEQ) was used to evaluate participants’ experiences of school bullying in the previous 12 months. It contains 16 items that are rated on a 4-point Likert scale [32,33]. This scale is composed of four 4-item subscales evaluating whether the respondent has experienced victimization from verbal or relational bullying (items 1–4, including social exclusion, being called a mean nickname, and being spoken ill of), physical bullying, or the snatching of belongings (items 5–8, including being beaten up, being forced to do work, or having money, school supplies, or snacks taken away). It also asks whether the respondent has perpetrated verbal or relational bullying (items 9–12), physical bullying, or the snatching of others’ belongings (items 13 to 16). The C-SBEQ was reported to have high internal consistency, with Cronbach’s alpha ranging from 0.727 to 0.753 [33].

#### 2.3.3. Suicidality

The 5-item questionnaire from the epidemiological version of the Kiddie Schedule for Affective Disorders and Schizophrenia [34] was used to assess the occurrence of suicidal ideation and attempts in the previous 12 months [35]. We used the Mandarin Chinese version of the aforementioned questionnaire, which was developed following rigorous methodological requirements regarding translation, back-translation, cultural adaptation, and psychometric properties [36]. Each question for suicide screening elicits a “yes” or “no” response. In a previous study, the Cohen’s kappa coefficient of agreement (κ) between participants’ self-reported suicide attempts and their parents’ reports was 0.541 (*p* < 0.001) [35]. A higher total number of items with the “yes” response indicates a higher risk of suicidality.

#### 2.3.4. Taiwanese Quality of Life Questionnaire for Adolescents

We used the self-reported Taiwanese Quality of Life Questionnaire for Adolescents (TQOLQA) to measure the participants’ QoL in the previous 12 months [37]. The 5-point, 38-item TQOLQA contains the following seven domains: family (7 items, e.g., “Are you satisfied with the relationship between you and your parents?”, Cronbach’s α = 0.91), residential environment (8 items, e.g., “Is your living environment healthy?”, Cronbach’s α = 0.88), personal competence (7 items, e.g., “How do you rate your learning ability?”, Cronbach’s α = 0.89), social relationships (5 items, e.g., “Do you have a friend who understands you well?”, Cronbach’s α = 0.80), physical appearance (4 items, e.g., “Do you feel uneasy about any part of your body or physical appearance?”, Cronbach’s α = 0.78), psychological well-being (4 items, e.g., “Does the feeling of depression or sadness interfere with your daily activities?”, Cronbach’s α = 0.79), and pain (3 items, e.g., “Does your pain or discomfort interfere with things you need to do?”, Cronbach’s α = 0.71). After the raw scores are converted for reverse questions, higher total scores indicate a higher QoL. The Cronbach’s α coefficient ranges from 0.77 to 0.91 for the global scale and seven domains [37].

### 2.4. Statistical Analysis

Statistical analyses were conducted using SAS 9.4 (SAS Institute Inc., Cary, NC, USA) and R 3.6.1 (R Core Team, Vienna, Austria). The descriptive results are expressed as frequencies and percentages for categorical variables, and as means and standard deviations (SD) for continuous variables.

To determine whether demographics, QoL, or the victimization and perpetration of school bullying differed between adolescents with ADHD with and without suicide-related problems, univariate analyses of odds ratios (ORs) with 95% confidence intervals (CIs) and independent *t* tests were conducted for binary and continuous variables respectively.

To investigate the relationships between ADHD symptoms, bullying, suicide, and QoL in more depth, we conducted a partial correlation network model analysis using the R package *qgraph* with “graphical least absolute shrinkage and selection operator (LASSO)” regularization with the hyperparameter set to 0.5 [38].

## 3. Results

Table 1 presents the demographics and mean scores of questionnaires for 203 participants with ADHD. The mean age (SD) of the participants was 14 (1.5) years. Most of the study population (78.8%) comprised boys. Overall, 19.7–22.7% reported suicidal ideation, 17.7% reported suicide planning, and 11.3% reported suicide attempts in the previous 12 months. Overall, 37.9% of participants reported suicide-related problems.

Significant differences were identified in gender, QoL, and victimization between adolescents with ADHD with and without suicide-related problems, as shown in Table 2. The proportion of suicide-related problems was significantly higher in girls with ADHD (63.6%) than in boys with ADHD (31.1%), with OR = 3.81 and 95% CI = 1.89–7.68. In addition, participants with suicide-related problems reported a lower QoL and higher victimization from school bullying, but no differences in age, ADHD symptoms, or perpetration of school bullying were noted.

We analyzed the correlation matrix to clarify the associations between school bullying, QoL, and suicidality among adolescents with ADHD, shown in Table 3. For the severity of ADHD, we observed that ADHD symptoms were associated with several QoL domains. Notably, inattention was positively associated with pain, whereas it was negatively associated with personal competence (Pearson’s *r* = −0.31). Hyperactivity/impulsivity, the other symptom of ADHD, was positively associated with personal competence and psychological well-being. School bullying was negatively associated with pain (*r* ranged from −0.13 to −0.27) and physical appearance (*r* ranged from −0.22 to −0.32). Suicidality was negatively associated with personal competence, psychological well-being (*r* = −0.63), physical appearance (*r* = −0.48), and the satisfaction of residential environment, whereas it was positively associated with social relationships.

To explore patterns of associations for types of school bullying with QoL and suicidality, we employed the partial correlation network model, as shown in Figure 1. Victimization from physical bullying correlated with personal competence, psychological well-being (QoL), and suicidality, shown in Figure 1A. However, we observed only associations with victimization, psychological well-being, and suicidality with verbal and relational bullying. A direct association between victimization and suicidality was lacking, shown in Figure 1B. Regarding the perpetration of school bullying, a similar triangular correlation was noted between the perpetration of physical bullying, personal competence, and suicidality, shown in Figure 1C. Perpetration and personal competence were not associated with verbal and relational bullying, shown in Figure 1D. In addition, ADHD symptoms had differing associations with personal competence and psychological well-being. Inattention was negatively associated with personal competence, whereas hyperactivity/impulsivity was positively associated with personal competence and psychological well-being.

## 4. Discussion

To the best of our knowledge, this is the first study to examine the associations between school bullying, QoL, suicidality, and ADHD symptoms simultaneously in adolescents. Our main finding is the triangular correlations between school bullying, QoL, and suicidality. These findings depict pathways from various types of school bullying to suicidality, where personal competence and psychological well-being may originate or mediate the pathways. Furthermore, ADHD symptoms may regulate these pathways. These findings are worthy of longitudinal investigations to clarify the causal relationships between school bullying, QoL, and suicidality, and the impacts of ADHD symptoms on these relationships.

We observed a strong triangular correlation among psychological well-being, victimization and suicidality. A possible mechanism for this is that victimization may decrease adolescents’ psychological well-being, thus promoting suicidality. The literature also supports the notion that psychological well-being plays a mediating role in the association between victimization and suicidality [39]. Our results can also be explained by the stress-diathesis model of suicide. In this model, an individual’s negative life events can cause stress that results in suicidal behaviors; however, the individual’s diathesis may regulate the effect of negative life events on suicide [40]. In the framing of our study, school bullying is a serious negative life event, and aspects of QoL represent an individual’s diathesis. If individuals have higher QoL, despite being involved in school bullying, they are less likely to develop suicidal behaviors. Furthermore, in addition to QoL, traits related to QoL have been suggested as protective diatheses against suicide, such as sense of purpose [41], self-determination [42], and resilience [40].

Personal competence was the other main QoL factor, with a triangular correlation with physical bullying and suicidality. The triangular correlation may originate from the bias of peers against perceived lower personal competence. This explanation is supported by a previous study, which demonstrated that adolescents are at a higher risk of being bullied if they have lower academic performance [43]. In addition, we only observed physical bullying that was directly involved in such a correlation, but not verbal-relational bullying. One possible reason is that physical bullying has a more profound and direct effect on intense suicidal behaviors, such as suicide attempts, than verbal and relational bullying in adolescents [10,44].

Lastly, we observed that the ADHD symptoms of inattention and hyperactivity/impulsivity may differentially regulate these triangular correlations. Inattention symptoms were negatively associated with personal competence as they indirectly increased suicidality through personal competence. By contrast, hyperactivity/impulsivity symptoms had an oppositional indirect association with suicidality through personal competence and a positive direct association with suicide. These differential patterns between QoL, suicide, and ADHD symptoms may be a result of differences in impairment caused by inattention and hyperactivity/impulsivity [45]. The ADHD inattentive subtype was reported to be more profoundly associated with academic problems, whereas the hyperactive-impulsive type was comparatively less impairing [46].

Finally, in our network model, we did not observe significant independent associations among ADHD symptoms, suicide, and school bullying. A plausible reason may be that all the study participants were patients with ADHD, resulting in a high homogeneity of ADHD symptoms, making the difference between individuals with and without suicide-related problems or school bullying issues difficult to identify. 

Future studies are warranted to clarify the regulating role of hyperactivity/impulsivity symptoms on these triangular correlations, as well as the correlations of separated items of the bullying scale to quality of life and suicidality.

### Limitations

The study had two limitations. Firstly, it was a cross-sectional study, so causal inferences for the relationship between school bullying, QoL, and suicidality were limited. Secondly, it used a retrospective survey, so our findings may have recall bias.

## 5. Conclusions

Several triangular correlations between types of school bullying, QoL, and suicidality were identified. The findings indicate possible pathways from several types of school bullying to suicidality, where personal competence and psychological well-being may originate or mediate the pathways. Moreover, ADHD symptoms may regulate these pathways. Longitudinal studies are required to clarify causal relationships in these triangular correlations.

## Figures and Tables

**Figure 1 ijerph-17-03262-f001:**
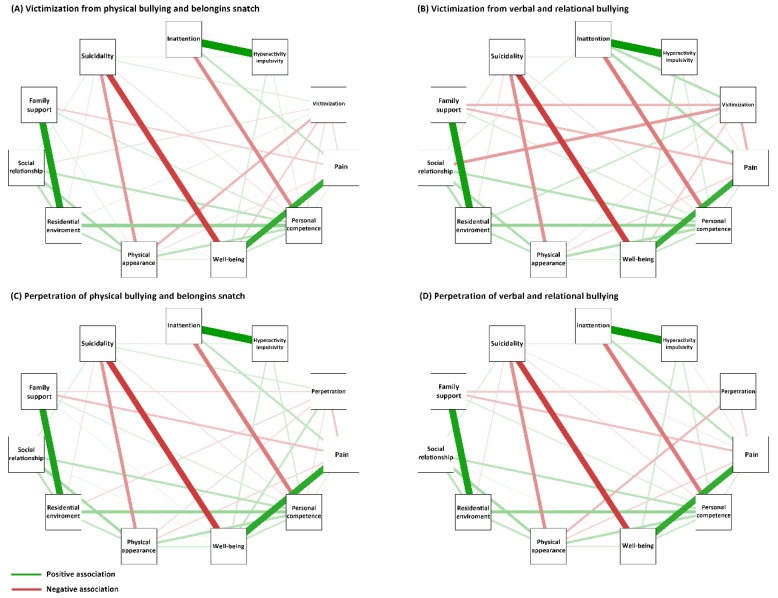
Associations between school bullying, quality of life, and suicidality among adolescents with ADHD. For victimization from physical bullying and belongings snatching, two triangular correlations were observed between personal competence, psychological well-being (QoL), and suicidality, shown in (**A**). However, the triangular correlation disappeared for verbal and relational bullying, shown in (**B**). A triangular correlation was also identified between the perpetration of physical bullying and belongings snatching, personal competence, and suicidality, shown in (**C**). The association between perpetration and personal competence was lacking for verbal and relational bullying, shown in (**D**). ADHD symptoms were associated with these triangular correlations. (Thickness of lines represents the intensity of association.)

**Table 1 ijerph-17-03262-t001:** Demographics, quality of life, school bullying, and suicide among adolescents with ADHD.

Variable	*N* = 203	Possible Range
Age, mean (SD)	14.0 (1.5)	-
Gender, *n* (%)		
Boy	160 (78.8%)	-
Girl	43 (21.2%)	-
Attention-deficit hyperactivity disorder, mean (SD)		
Inattention	15.7 (5.6)	[0, 27]
Hyperactivity/impulsivity	9.6 (6.3)	[0, 27]
Quality of life, mean (SD)		
Pain	11.8 (2.1)	[3, 15]
Personal competence	20.3 (4.9)	[7, 35]
Psychological well-being	15.1 (3.3)	[4, 20]
Physical appearance	15.5 (3.2)	[4, 20]
Residential environment	28.2 (6.1)	[8, 40]
Social relationship	17.6 (4.0)	[5, 25]
Family	23.4 (5.4)	[7, 35]
Total score	131.7 (20.2)	[38, 190]
Bullying involvement, mean (SD)		
Victimization	3.3 (3.2)	[0, 24]
Victim of verbal and relational bullying	2.5 (2.6)	[0, 12]
Victim of physical bullying and belongings snatching	0.9 (1.4)	[0, 12]
Perpetration	2.9 (2.7)	[0, 24]
Perpetrator of verbal and relational bullying	2.2 (2.1)	[0, 12]
Perpetrator of physical bullying and belongings snatching	0.6 (1.0)	[0, 12]
Suicide, *n* (%)		
Suicidal ideation	46 (22.7)	-
Desire to die	40 (19.7)	-
Thought of suicide attempt	41 (20.2)	-
Suicide plan	36 (17.7)	-
Suicide attempt	23 (11.3)	-
At least one suicidal response	78 (38.2)	-

Possible ranges of items of ADHD symptoms, QoL, bullying involvement were presented as [minimum, maximum].

**Table 2 ijerph-17-03262-t002:** Differences in demographics, quality of life, and school bullying between adolescents with ADHD with and without suicidality.

Variable	Suicidality	Non-Suicidality	Statistic
	*N* = 78	*N* = 125	
Variable	*n (%)*	*n (%)*	odds ratio (95% CI)
Gender			
Boy	50 (31.1)	109 (68.6)	1.00
Girl	28 (63.6)	16 (36.4)	3.81 (1.89–7.68) ***
	*Mean (SD)*	*Mean (SD)*	*t and p value*
Age	14.0 (1.5)	14.0 (1.4)	*t* = 0.13, *p* = 0.892
Attention-deficit hyperactivity disorder			
Inattention	15.8 (5.6)	15.6 (5.7)	*t* = −0.27, *p* = 0.785
Hyperactivity/impulsivity	9.7 (6.4)	9.4 (6.3)	*t* = −0.33, *p* = 0.746
Quality of life	121.2 (18.6)	138.0 (18.4)	*t* = 6.26, *p* < 0.001 ***
Victimization	4.4 (3.5)	2.6 (2.9)	*t* = −3.69, *p* < 0.001 ***
Verbal and relational bullying	3.1 (2.7)	2.0 (2.4)	*t* = −2.94, *p* = 0.004 **
Physical bullying	1.3 (1.5)	0.6 (1.1)	*t* = −3.45, *p* < 0.001 ***
Perpetration	3.2 (2.6)	2.7 (2.8)	*t* = −1.21, *p* = 0.226
Verbal and relational bullying	2.4 (2.0)	2.2 (2.2)	*t* = −0.75, *p* = 0.456
Physical bullying	0.8 (1.1)	0.5 (1.0)	*t* = −1.52, *p* = 0.129

** *p* < 0.01, *** *p* < 0.001.

**Table 3 ijerph-17-03262-t003:** Correlation matrix of ADHD symptoms, school bullying, suicide, and quality of life.

Variable	ADHD		Bullying Victimization	Bullying Perpetration			Suicide	Quality of Life						
	1	2	3	4	5	6	7	8	9	10	11	12	13	14
1. Inattention	1.00													
2. Hyperactivity/impulsivity	0.60	1.00												
3. Verbal and relational bullying	0.19	0.11	1.00											
4. Physical bullying	0.08	0.04	0.43	1.00										
5. Verbal and relational bullying	0.04	0.01	0.29	0.21	1.00									
6. Physical bullying	0.06	0.07	0.21	0.27	0.44	1.00								
7. Suicide	0.07	0.04	0.24	0.32	0.10	0.20	1.00							
8. Pain	0.15	0.09	−0.27	−0.26	−0.13	−0.13	−0.37	1.00						
9. Personal competence	−0.31	−0.10	−0.21	−0.27	−0.13	−0.25	−0.40	0.21	1.00					
10. Psychological well-being	0.02	0.09	−0.31	−0.34	−0.13	−0.08	−0.63	0.59	0.41	1.00				
11. Physical appearance	−0.13	−0.09	−0.25	−0.32	−0.22	−0.24	−0.48	0.14	0.49	0.39	1.00			
12. Residential environment	−0.11	−0.04	−0.15	−0.12	−0.12	−0.28	−0.35	0.12	0.55	0.32	0.48	1.00		
13. Social relationship	−0.08	−0.08	−0.32	−0.21	−0.06	−0.16	−0.20	0.13	0.40	0.25	0.40	0.41	1.00	
14. Family	−0.12	−0.08	−0.22	−0.04	−0.17	−0.25	−0.21	−0.02	0.42	0.20	0.36	0.66	0.30	1.00

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
