# Peer review of "Correlations between Quality of Life, School Bullying, and Suicide in Adolescents with Attention-Deficit Hyperactivity Disorder"

_ijerph, 2020, doi:10.3390/ijerph17093262_

Round 1

Reviewer 1 Report

First and foremost, I want to thank the Journal to give me the opportunity to serve as a reviewer in general, and specifically, I want to thank the authors for giving me the chance to provide feedback on this manuscript, which was interesting and easy to read, so I really liked it.

I applaud the authors for taking on research on bullying and suicidality in adolescents, as it is an important issue. I also applaud the authors for their approach to consider victimization and perpetration concurrently. And last but not least, I do applaud the authors for considering potential moderators that could change the association between bullying and suicidality.

However, I do NOT see how it can be conceptually argued, or empirically concluded from the data presented, that quality of life acts as a moderator variable “buffering” the effect of bullying on suicidality. Quality of life is an umbrella construct that captures the subjective satisfaction with different aspects of one’s life, and it necessarily covaries with thoughts about ending this life (the DV in the model presented), with the experience of bullying (the IV in the model presented), not to mention how strongly it must be shaped by depression (a third variable currently unassessed), one of the main reasons for suicidality.

However, in the manuscript, quality of life scores are discussed as if they had no substantial overlap with the independent and dependent variable. Instead, quality of life is discussed as if it was an objective indication of resources or liabilities that can be separated from bullying and suicidality.

The reported interaction effect in a regression model (that is underpowered for the reported 3way interactions) can NOT be taken as proof for quality of life intervening in the path from victimization to suicidality. Thus, the conclusions stated in the manuscript, even in its title, are not warranted based on the presented data and analyses, e.g. “Due to the buffering effect of quality of life on the relationship between bullying victimization and suicidality in adolescents with ADHD, it is valuable to establish an effective program by incorporating quality of life to prevent suicide from school bullying in the future.”

Discussing the results as opening up new potential for preventing suicide is misleading, as if ‘bullying-induced suicidality’ can be prevented by improving quality of life by x points, no matter on what subscale of the scale this improvement is achieved.

I hate to feel the need to be so clear, but I do not recommend publication of the paper with the current analyses and interpretation of the data. I do, however, see quite some potential in the data, the research question, and large parts of the manuscript, as it is interesting to know about the associations of different types of bullying, including perpetration, and the different subscales of quality of life, and suicidality, in people with ADHD.  

So what I suggest is a rather big revision:

Do not attempt to insinuate any causal relationships, neither by making use of regression models nor by making use of words such as ‘buffering effect’ or ‘incorporating QoL to prevent suicide’. Instead of regression models, report a NETWORK MODEL (or, if that is absolutely unfeasible for reasons that must be outlined, a comparable analysis) describing the associations of all of the subscales of bullying and Quality of Life, as well as suicidality, and, if available, severity of ADHD symptoms. It will be really enlightening to interested readers and practitioners to learn which subscales of QoL may be particularly suitable as flagging impending suicidality, and which subscales of QoL are most affected by bullying. I am not sure whether power will be sufficient for this.

In addition, I would love to see (but abstain from making this a requirement for acceptance) more data: First, a bigger sample of ADHD individuals would be interesting in general, as the associations hinted at here are more than overwhelming for such a small sample. Second, it deems me highly interesting to recruit a comparison group of adolescents without ADHD that are comparable to the previous sample on most other characteristics other than their ADHD status.

Being able to compare the associations of bullying, quality of life and suicidality between ADHD and non-ADHD adolescents, or ADHD symptom type or strength with bullying, QoL and suicidality, definitely would increase the merit of the research, and I would love to know more about this. Recruiting a comparison group would also solve another issue:

So far, the choice of ADHD patients for the research question has not become clear to me and definitely needs to be motivated better, which would become clearer if it aimed at identifying the specific strain that ADHD may put on adolescents.

On another note: While the manuscript is generally extremely well-written in terms of formulations (thanks, and congratulations!), it suffers from a magnitude of extraordinary language errors.

I am genuinely confused how these types of errors may have emerged in a manuscript with an otherwise excellent level of English. An explanation might be the use of a software tool that automatically improves writing style - which is fine, but please, have an actual person with excellent command of academic English proof-read the manuscript and get rid of these mistakes which are more than a nuisance (like typos) but in some cases distort the (assumed intended) meaning.

I hope my major change suggestions do not blur the fact that I see a lot of merit in the manuscript’s research question, and the manuscript’s readability. I would love to see the network model put into practice, and see an improved version of the manuscript being published soon.

Author Response

Responses to Reviewers’ Comments

Ms. No.: ijerph-764254

Article Title: Quality of Life Buffers the Impact of Victimization and Perpetration of School Bullying on Suicide in Adolescents with Attention-Deficit/Hyperactivity Disorder

Comments from the Editors and/or Reviewers:

Reviewer #1:

Comments to the Author

First and foremost, I want to thank the Journal to give me the opportunity to serve as a reviewer in general, and specifically, I want to thank the authors for giving me the chance to provide feedback on this manuscript, which was interesting and easy to read, so I really liked it.

I applaud the authors for taking on research on bullying and suicidality in adolescents, as it is an important issue. I also applaud the authors for their approach to consider victimization and perpetration concurrently. And last but not least, I do applaud the authors for considering potential moderators that could change the association between bullying and suicidality.

1-      However, I do NOT see how it can be conceptually argued, or empirically concluded from the data presented, that quality of life acts as a moderator variable “buffering” the effect of bullying on suicidality. Quality of life is an umbrella construct that captures the subjective satisfaction with different aspects of one’s life, and it necessarily covaries with thoughts about ending this life (the DV in the model presented), with the experience of bullying (the IV in the model presented), not to mention how strongly it must be shaped by depression (a third variable currently unassessed), one of the main reasons for suicidality.

However, in the manuscript, quality of life scores are discussed as if they had no substantial overlap with the independent and dependent variable. Instead, quality of life is discussed as if it was an objective indication of resources or liabilities that can be separated from bullying and suicidality.

Reply: At the reviewer’s suggestion, we performed a network model analysis to present the relationships among ADHD, quality of life (QoL), school bullying, and suicide. We do not discuss the moderation effect of QoL; instead, we explain our results from a broader perspective. Because of the change of analysis strategies, we have substantially rewritten our discussion. Please refer to our Discussion section.

2-   The reported interaction effect in a regression model (that is underpowered for the reported 3way interactions) can NOT be taken as proof for quality of life intervening in the path from victimization to suicidality. Thus, the conclusions stated in the manuscript, even in its title, are not warranted based on the presented data and analyses, e.g. “Due to the buffering effect of quality of life on the relationship between bullying victimization and suicidality in adolescents with ADHD, it is valuable to establish an effective program by incorporating quality of life to prevent suicide from school bullying in the future.”

Discussing the results as opening up new potential for preventing suicide is misleading, as if ‘bullying-induced suicidality’ can be prevented by improving quality of life by x points, no matter on what subscale of the scale this improvement is achieved.

Reply: We have withdrawn the moderation analysis and discussion of the buffering effect accordingly. In addition, the discussion of an intervention program based on QoL has been removed.

3-   Discussing the results as opening up new potential for preventing suicide is misleading, as if ‘bullying-induced suicidality’ can be prevented by improving quality of life by x points, no matter on what subscale of the scale this improvement is achieved.

Reply: As mentioned in the preceding reply, the discussion of the QoL intervention program and its possible effects has been removed.

4-      I hate to feel the need to be so clear, but I do not recommend publication of the paper with the current analyses and interpretation of the data. I do, however, see quite some potential in the data, the research question, and large parts of the manuscript, as it is interesting to know about the associations of different types of bullying, including perpetration, and the different subscales of quality of life, and suicidality, in people with ADHD. So what I suggest is a rather big revision: Do not attempt to insinuate any causal relationships, neither by making use of regression models nor by making use of words such as ‘buffering effect’ or ‘incorporating QoL to prevent suicide’. Instead of regression models, report a NETWORK MODEL (or, if that is absolutely unfeasible for reasons that must be outlined, a comparable analysis) describing the associations of all of the subscales of bullying and Quality of Life, as well as suicidality, and, if available, severity of ADHD symptoms. It will be really enlightening to interested readers and practitioners to learn which subscales of QoL may be particularly suitable as flagging impending suicidality, and which subscales of QoL are most affected by bullying. I am not sure whether power will be sufficient for this.

Reply:

  • We followed the reviewer’s suggestion and performed network model analyses to determine the relationships among ADHD, QoL, and suicide in different types of bullying, as visualized in Figures 1A–1D. We have accordingly revised the Discussion section substantially. These changes have been added in the Methods, Results and Discussion section as follows:

Statistical Analysis (lines 198-202 of the manuscript of clean version)

To investigate relationships between ADHD symptoms, bullying, suicide, and QoL in more depth, we conducted a partial correlation network model analysis using the R package qgraph with “graphical LASSO” regularization with the hyperparameter set to 0.5 [37].

Results (lines 236-249)

To explore patterns of associations for types of school bullying with QoL and suicidality, we employed the partial correlation network model (Figure 1). Victimization from physical bullying correlated with personal competence and psychological well-being (QoL) and suicidality one another (Figure 1A). However, we observed only associations of victimization, psychological well-being, and suicidality with verbal and relational bullying; a direct association between victimization and suicidality was lacking (Figure 1B). Regarding the perpetration of school bullying, a similar triangular correlation was noted between perpetration of physical bullying, personal competence, and suicidality (Figure 1C). Perpetration and personal competence were not associated with verbal and relational bullying (Figure 1D). In addition, ADHD symptoms had differing associations with personal competence and psychological well-being. Inattention was negatively associated with personal competence, whereas hyperactivity/impulsivity was positively associated with personal competence and psychological well-being.

Figure 1

  • We have deleted causal terms, such as effect and prevention, from the Discussion section of this manuscript.

6-      In addition, I would love to see (but abstain from making this a requirement for acceptance) more data: First, a bigger sample of ADHD individuals would be interesting in general, as the associations hinted at here are more than overwhelming for such a small sample. Second, it deems me highly interesting to recruit a comparison group of adolescents without ADHD that are comparable to the previous sample on most other characteristics other than their ADHD status.

Being able to compare the associations of bullying, quality of life and suicidality between ADHD and non-ADHD adolescents, or ADHD symptom type or strength with bullying, QoL and suicidality, definitely would increase the merit of the research, and I would love to know more about this. Recruiting a comparison group would also solve another issue:

Reply: We are not able to increase our sample size, and we did not recruit adolescents without ADHD in our sample. However, we conducted a network analysis, based on the reviewer’s suggestion, stratified by the type of school bullying to increase the merit of the research. Please refer to the revised Methods section.

7- So far, the choice of ADHD patients for the research question has not become clear to me and definitely needs to be motivated better, which would become clearer if it aimed at identifying the specific strain that ADHD may put on adolescents.

Reply: We have changed our analysis strategy to network analysis and incorporated ADHD symptoms. We found triangular correlations among school bullying, QoL, and suicidality. Moreover, ADHD symptoms may regulate these pathways. We hope that these changes to our manuscript clarify the role of ADHD symptoms. These changes have been added in the Results and Discussion section as follows:

Results (lines 246-249)

In addition, ADHD symptoms had differing associations with personal competence and psychological well-being. Inattention was negatively associated with personal competence, whereas hyperactivity/impulsivity was positively associated with personal competence and psychological well-being.

Discussion (lines 312-322)

Lastly, we observed that the ADHD symptoms of inattention and hyperactivity/impulsivity may differentially regulate these triangular correlations. Inattention symptoms were negatively associated with personal competence; they indirectly increased suicidality through personal competence. By contrast, hyperactivity/impulsivity symptoms had an oppositional indirect association with suicidality through personal competence and had a positive direct association with suicide. These differential patterns between QoL, suicide, and ADHD symptoms may be a result of differences in impairment caused by inattention and hyperactivity/impulsivity [46]. The ADHD inattentive subtype was reported to be more profoundly associated with academic problems, whereas the hyperactive-impulsive type was comparatively less impairing [47].

8-      On another note: While the manuscript is generally extremely well-written in terms of formulations (thanks, and congratulations!), it suffers from a magnitude of extraordinary language errors.

I am genuinely confused how these types of errors may have emerged in a manuscript with an otherwise excellent level of English. An explanation might be the use of a software tool that automatically improves writing style - which is fine, but please, have an actual person with excellent command of academic English proof-read the manuscript and get rid of these mistakes which are more than a nuisance (like typos) but in some cases distort the (assumed intended) meaning.

Reply: We apologize for such errors. They may have emerged in our manuscript because we are not native speakers of English and did not use translation software. We personally wrote every word in this manuscript. We have sent this manuscript for English editing to improve the readability.

I hope my major change suggestions do not blur the fact that I see a lot of merit in the manuscript’s research question, and the manuscript’s readability. I would love to see the network model put into practice, and see an improved version of the manuscript being published soon.

Reply: We genuinely appreciate the reviewer’s valuable suggestions. We believe that the changes made to this manuscript based on these suggestions make it complete.

Reviewer 2 Report

Thank you for the opportunity to review your research. There are a few areas that I think would benefit from revision. 

  1. The data is now somewhat old being from 2009-2012 - why is it now being presented for publication? Is this a further analysis of already published data? I see a 2019 (on line first 2017) publication from your group which appears to be the same data set? That article appears to make reference to several other publications from the group. Given both the age of the data and the apparent other publications from the data set, I think we need some explanation around this. There is a point where we try to draw too much from our data so the journal will need to better understand this issue.
  2. The work uses the WHO basis for discussing quality of life although given the focus area of this work there might be some merit in broader considerations such as sense of purpose, relational strengths as buffers as well as resiliance. Recent research has suggested that these are protective factors (see for example Blazek et al., 2015 and Bureau et al., 2012).
  3. Lines 34-35 - the meaning here is a bit confusing so a rewrite of the sentence would be useful.
  4. Line 80 - including self-injurious behaviours here is a bit concerning given the body of research that helps us to see that behaviour as quite distinct from suicidal behaviours.
  5. Line 115 - suppose - is this a hypothesis?
  6. Line 151 - Did parents consent?
  7. Line 167 - The Kiddie Schedule - can you let us know about the cultural applicability of this interview schedule?
  8. Line 195 - It is not clear how severity was assessed and measured.
  9. Lines 313-326 - are largely repeats of introduction - this can be edited down noting the earlier mentions and overview.
  10. Line 327 - ?snatch - what are you meaning here?
  11. Line 335 - to several diseases - to doesn't work here.
  12. Limitations - I think it is worth considering that the data collection is based upon retrospective views which can cause recall bias for example.

Author Response

Reviewer #2:

Comments to the Author

Thank you for the opportunity to review your research. There are a few areas that I think would benefit from revision.

1.      The data is now somewhat old being from 2009-2012 - why is it now being presented for publication? Is this a further analysis of already published data? I see a 2019 (on line first 2017) publication from your group which appears to be the same data set? That article appears to make reference to several other publications from the group. Given both the age of the data and the apparent other publications from the data set, I think we need some explanation around this. There is a point where we try to draw too much from our data so the journal will need to better understand this issue.

Reply: Although the data were collected from 2009 to 2012, we did not have enough time for data cleaning and analysis until recently because of our heavy workload. The first paper using these data was published in 2019, and we have clearly indicated this information in the Participants section as follows:

Participants (lines 122-130 of the manuscript of clean version)

Originally, a total of 579 children and adolescents diagnosed as having ADHD were invited to participate. Among them, 474 (81.9%) agreed to participate and were interviewed by research assistants using the study questionnaire. A study based on the same data has been published [4]. In this study, we included participants aged ≥12 years because those younger than 12 did not complete the assessment of QoL, one of the variables of interest. After excluding these participants, a total of 203 adolescents diagnosed with ADHD aged ≥12 years were included. No significant differences were identified in gender (p > .05) or age (p > .05) between those who agreed and those who refused to participate.

Reference

Yeh, Y. C., Huang, M. F., Wu, Y. Y., Hu, H. F., & Yen, C. F. (2019). Pain, bullying involvement, and mental health problems among children and adolescents with ADHD in Taiwan. Journal of Attention Disorders, 23(8), 809-816.

2. The work uses the WHO basis for discussing quality of life although given the focus area of this work there might be some merit in broader considerations such as sense of purpose, relational strengths as buffers as well as resiliance. Recent research has suggested that these are protective factors (see for example Blazek et al., 2015 and Bureau et al., 2012).

Reply: We have added discussion of these merits to our Discussion section as follows:

Discussion (lines 309-311)

Furthermore, in addition to QoL, traits related to QoL have been suggested as protective diatheses against suicide, such as sense of purpose [44], self-determination [45], and resilience [43].

3.      Lines 34-35 - the meaning here is a bit confusing so a rewrite of the sentence would be useful.

Reply: Following the other reviewer’s suggestion, we have changed our analysis strategy, and these sentences have been modified and rewritten in the Abstract as follows:

Abstract (lines 28-31)

Although adolescents with attention‐deficit hyperactivity disorder (ADHD) have a higher risk of suicidality and more problems related to school bullying, and quality of life (QoL) has been reported to associated with school bullying, suicide, and ADHD, no study has examined their correlations.

4.     Line 80 - including self-injurious behaviours here is a bit concerning given the body of research that helps us to see that behaviour as quite distinct from suicidal behaviours.  

Reply: We have removed “self-injurious behaviors” from the sentence in the Introduction.

Introduction (lines 67-70)

Similar results regarding victimization have been found in longitudinal studies, in which researchers reported that bullying victimization at an early age could predict suicidal ideation and suicide attempts in adolescents

5.      Line 115 - suppose - is this a hypothesis?

Reply: We have deleted this sentence. All hypotheses are presented in the last paragraph of the Introduction section as follows:

Introduction (lines 105-108)

Based on the aforementioned literature, we hypothesize that adolescents with experiences of school bullying report lower QoL, and that a negative association is observed between QoL and suicidality. We also examined the roles of ADHD symptoms in these associations.

6.      Line 151 - Did parents consent?

Reply: Yes, we have added the information to the Participants section as follows:

Ethics (lines 134-136)

All participants and their parents provided written informed consent before completing the questionnaires.

7.      Line 167 - The Kiddie Schedule - can you let us know about the cultural applicability of this interview schedule?

Reply: The Kiddie Schedule for Affective Disorders and Schizophrenia has demonstrated reliable cultural applicability across major ethnic groups and cultures (Greenbaum et al., 2017). We developed a Mandarin version of the Kiddie Schedule for Affective Disorders and Schizophrenia with rigorous methodological requirements in translation, back-translation, cultural adaptation, and psychometrics (Chen et al., 2017). This information and related references have been added to the Methods section as follows:

Suicidality (lines 163-166)

We used the Mandarin Chinese version of the aforementioned questionnaire, which was developed following rigorous methodological requirements regarding translation, back-translation, cultural adaptation, and psychometric properties [35].

References

Greenbaum, B., Rosenberg, M. & Dalenberg, C. (2017). Schedule for affective disorders and schizophrenia. In A. Wenzel (Ed.), The sage encyclopedia of abnormal and clinical psychology (Vol. 1, pp. 2917-2919). Thousand Oaks, CA: SAGE Publications, Inc. doi: 10.4135/9781483365817.n1171

Chen, Y. L., Shen, L. J., & Gau, S. S. (2017). The mandarin version of the Kiddie-Schedule for Affective Disorders and Schizophrenia-Epidemiological version for DSM-5 - a psychometric study. Journal of the Formosan Medical Association, 116(9), 671-678. doi:10.1016/j.jfma.2017.06.013

8.      Line 195 - It is not clear how severity was assessed and measured.

Reply: We have described the SNAP-IV to the Methods section to state how the severity of ADHD symptoms was measured and assessed as follows:

Measures ((lines 138-146)

Chinese version of the Swanson, Nolan, and Pelham, Version IV

The Swanson, Nolan, and Pelham, Version IV (SNAP-IV) employs a 4-point Likert scale with a score of 0 (not at all), 1 (just a little), 2 (quite a bit), and 3 (very much). It contains 18 items according to the core symptoms identified in the DSM-IV for symptoms of inattention-deficit (items 1–9) and hyperactivity/impulsivity (items 10–18) [29]. We used the Chinese version of SNAP-IV parent forms to assess ADHD-related symptoms of participants; the version has demonstrated satisfactory test–retest reliability, internal consistency, and concurrent and discriminant validity [30].

9.      Lines 313-326 - are largely repeats of introduction - this can be edited down noting the earlier mentions and overview.

Reply: Because of the new analysis, we have substantially modified the paragraph to avoid repetition, and many sentences have been deleted. The original paragraph was the discussion for moderation analysis, which has been removed. A new paragraph was added to discuss a network model analysis based on the reviewer's suggestion.

10.      Line 327 - ?snatch - what are you meaning here?

Reply: Because of the new analysis, we have deleted this paragraph.

11.      Line 335 - to several diseases - to doesn't work here.

Reply: The intervention of QoL has been removed based on the reviewer's suggestion. Thus, we have deleted this sentence and corresponding paragraph.

12.      Limitations - I think it is worth considering that the data collection is based upon retrospective views which can cause recall bias for example.

Reply: We have added the use of retrospective data to the considered limitations as follows:

Limitations (line 334)

Second, it used a retrospective survey, and our findings may have recall bias.

Round 2

Reviewer 1 Report

I applaud the authors for so quickly implementing that many changes in their manuscript and agree with their belief that „the changes made to this manuscript based on these suggestions make it complete.“ 

I am glad to be able to conclude that in this version, the merit to the scientific community is given and substantial, and I recommend publication in the International Journal of Environmental Research and Public Health. 

I do, however, still have some minor suggestions that I would like the authors to take into consideration: 

1. Be more specific about your main findings in both abstract and the conclusion section - this is your main merit and it should be visible at first glance 

2. In the table with the descriptives, these may be of greater informative value if accompanied by the range of possible values, so that the mean and standard deviation can directly be interpreted as big/small etc.

3. Table 3 carries great information; however, each variable should be named within the table, despite the obvious size problem which I understand, but this can be solved by using small fonts or (easy to understand) abbreviations. Otherwise, it is really hard to extract the information so valuable.

4. The use of „x correlates with y1 and y2 and y3 one another“ is wrong and confusing, thus, in lines like 242 and 267, „one another“ needs to be deleted.

5. The figure of the network analysis: I suggest integrating all 4 graphs in one figure, as this also allows to see the relations between being a victim and a perpetrator of bullying, and the relations between the different types of bullying. If this does not deem you possible, please at least make it very visibly clear that the only that changes between the four graphs is the bullying-related variable on the upper right side, highlighting it, as otherwise it takes a lot of time to figure out what the difference between the graphs is, and a voluntary reader may quickly abandon that prospect, and much of your research’s merit and impact may be lost unduly. 

6. When discussing your findings, just for parsimony, I suggest you start off with psychological well-being first, as this has a much stronger associations with suicidality, and then move on to discuss personal competence.

(7. For a future study (or, if you want to, also in the present one, could also be supplemental materials), the impact of separate items of the bullying subscales in their relation to quality of life and suicidality deems highly interesting, also when comparing an adhd and a non-adhd sample. For instance, social exclusion has often been discussed as „worse“ than negative attention, but as it is collapsed here in the subscale of verbal and relational aggression, this cannot be checked for by the reader.)

Thanks for being so responsive and this valuable addition to our understanding of the relation of quality of life, bullying and suicidality in ADHD youth.

Author Response

We appreciate your valuable comments. Attached please find the reply.

Reviewer 2 Report

Thank you for the revised version. As can happen with revisions, new issues arise. I have noted them in the version using the review / track changes function in word.

Perhaps my significant concern is in the discussion. In my view, the data is narrow in its value and should be reported as such. There is data that was not collected but the discussion begins to speculate what meaning might be there - it is not and that detracts from what the research can tell us. 

The new figures are very helpful - they might need a little bit more description so that readers who are often drawn to visual material can easily digest meaning. 

Author Response

We appreciate your valuable comments. Attached please find our reply.

Round 3

Reviewer 2 Report

Thank you for your revisions and further analysis. The article is much stronger now. In my view, it is ready for publication